# Development of Oleogel-in-Water High Internal Phase Emulsions with Improved Physicochemical Stability and Their Application in Mayonnaise

**DOI:** 10.3390/foods13172738

**Published:** 2024-08-29

**Authors:** Jingjing Yu, Mingyue Yun, Jia Li, Yanxiang Gao, Like Mao

**Affiliations:** 1Key Laboratory of Healthy Beverages, China National Light Industry, College of Food Science and Nutritional Engineering, China Agricultural University, Beijing 100083, China; 2018306100711@cau.edu.cn (J.Y.); yunmingyue1999@163.com (M.Y.); lijia207@gmail.com (J.L.); drgyx@sina.com (Y.G.); 2CAU Sichuan Chengdu Advanced Agricultural Industrial Institute, Chengdu 611430, China

**Keywords:** high internal phase emulsions, whey protein isolate, mayonnaise, rheological properties, stability

## Abstract

Egg-free mayonnaise is receiving greater attention due to its potential health benefits. This study used whey protein isolate (WPI) as an emulsifier to develop high internal phase emulsions (HIPEs) based on beeswax (BW) oleogels through a simple one-step method. The effects of WPI, NaCl and sucrose on the physicochemical properties of HIPEs were investigated. A novel simulated mayonnaise was then prepared and characterized. Microstructural observation revealed that WPI enveloped oil droplets at the interface, forming a typical O/W emulsion. Increase in WPI content led to significantly enhanced stability of HIPEs, and HIPEs with 5% WPI had the smallest particle size (11.9 ± 0.18 μm). With the increase in NaCl concentration, particle size was increased and ζ-potential was decreased. Higher sucrose content led to reduced particle size and ζ-potential, and slightly improved stability. Rheological tests indicated solid-like properties and shear-thinning behaviors in all HIPEs. The addition of WPI and sucrose improved the structures and viscosity of HIPEs. Simulated mayonnaises (WE-0.3%, WE-1% and YE) were then prepared based on the above HIPEs. Compared to commercial mayonnaises, the mayonnaises based on HIPEs exhibited higher viscoelastic modulus and similar tribological characteristics, indicating the potential application feasibility of oleogel-based HIPEs in mayonnaise. These findings provided insights into the development of novel and healthier mayonnaise alternatives.

## 1. Introduction

Mayonnaise, a versatile and widely consumed condiment, is a typical oil-in-water (O/W) emulsion, having an oil content in the range of 70–80% [1]. Traditional mayonnaise is known for its creamy texture, smooth mouthfeel, and ability to enhance the flavor of various foods. Egg yolk is the main ingredient in mayonnaise, prized for its excellent emulsifying properties due to its high lecithin and protein contents. While egg yolk is highly nutritious and offers many health benefits, concerns arise when it is consumed in excessive quantities, due to the high level of cholesterol and saturated fats [2]. To cater to specific dietary needs, one effective strategy is to develop low-cholesterol high internal phase emulsions (HIPEs) by replacing egg yolk with food-grade emulsifiers and stabilizers [3]. HIPEs, known for their high viscosity and semi-solid structures [4], can mimic the rheological properties, tribological properties, and overall sensory experience of traditional mayonnaise [5,6].

HIPEs refer to emulsions with a dispersed phase volume higher than 74%. HIPEs have high viscoelasticity and gel-like properties, because of the dense packing of the dispersed droplets. As reported in the literature, HIPEs have been widely used in the fields of functional foods, materials, and cosmetics [7]. Relevant research showed that protein, polysaccharide and other natural biological macromolecules could replace egg yolk as emulsifiers and thickeners in mayonnaise-type emulsions, which had a positive impact on stability and rheological properties [8]. Andrêssa et al. [9] prepared HIPEs with lentil protein isolate, and these HIPEs presented a creamy texture and elastic behavior, and could be utilized as promising alternatives to saturated fat. However, the large interfacial area makes it hard to prepare stable HIPEs for widespread food application. Additionally, protein-based emulsifiers are susceptible to environmental factors, like temperature fluctuations, ionic strength variations, and pH levels [10]. According to literature research, solidifying or gelling the oil phase served as a useful strategy for enhancing emulsion stability [11]. Notably, gelation of the oil phase (oleogelation) offered a viable solution without compromising the chemical integrity of liquid vegetable oil, thus preserving its healthy attributes [12]. Oleogelation could entrap liquid oil inside a three-dimensional network structure facilitated by Van der Waals force, electrostatic attraction and other interactions. Beeswax (BW) was a natural small molecule oleogelator, and showed a strong ability to convert liquid oils into a semi-solid gel form [13]. In emulsion systems, adding BW could increase the oil phase density, thereby minimizing density disparities with the aqueous phase and enhancing gravitational stability [14]. In the study of Zhang et al. [15], addition of BW oleogels created a synergistic effect with whey protein isolate (WPI), promoting the formation of stable emulsions. Flaxseed oil had a high content of omega-3 fatty acids, which could provide health benefits, such as lowering cholesterol and preventing diabetes and hypertension [16]. It could be used as a substitute for traditional oils in mayonnaise to enhance the mayonnaise’s nutritional value.

Traditionally, mayonnaise is prepared by mixing egg yolk, water, salt, edible oil, vinegar, sucrose, and additional ingredients to develop an oil-in-water emulsion. WPI has good amphiphilic nature and the potential to replace egg yolk in the preparation of simulated mayonnaise [17]. WPI is a natural biological macromolecule emulsifier that is non-toxic to the human body and meets current consumers’ demands for clean labels. Derived from whey, a byproduct of cheese production, WPI is mainly composed of β-lactoglobulin and α-lactalbumin [18]. WPI molecules adsorb onto the surface of oil droplets, forming a protective layer that prevents coalescence and maintains the stability of the emulsion [19]. This stabilizing effect is due to the ability of WPI to reduce interfacial tension and hinder oil droplets’ movement within the aqueous phase. Salt and sugar also play crucial roles in the formulation of mayonnaise, influencing both its taste and stability. Salt not only enhances flavor but also contributes to the preservation of mayonnaise by inhibiting microbial growth. Additionally, salt affects the rheological properties of mayonnaise, influencing its texture and viscosity [20]. On the other hand, sugar contributes to flavor balance and sweetness perception while also impacting the stability of mayonnaise by interacting with other ingredients and affecting the emulsification process [21]. It is necessary to explore the influence of mayonnaise ingredients, including emulsifier, NaCl, and sucrose, in order to achieve the desired sensory attributes and stability for the relevant products.

This study focused on the development of oleogel-in-water HIPEs stabilized by WPI and beeswax oleogel. By adjusting the content of WPI, NaCl and sucrose, the microstructures, physicochemical stability and rheological properties of HIPEs were evaluated. Subsequently, novel mayonnaises were prepared based on the HIPEs with gelled oil phase, and their application qualities were characterized. In addition, two commercial mayonnaise products were used as references. Compared with commercial mayonnaises, the feasibility of applying oleogel-based HIPEs in mayonnaise was studied based on the aspects of appearance, microstructures, rheological properties and tribological properties. The acquired knowledge is anticipated to offer a theoretical foundation for application and further investigation of HIPEs in healthier mayonnaise products.

## 2. Materials and Methods

### 2.1. Materials

Whey protein isolate (WPI) was purchased from Davisco Foods International, INC. (Le Sueur, MN, USA). Flaxseed oil (Inner Mongolia Glynol Biology Co., Ltd., Inner Mongolia, Baotou, China) was bought from a local supermarket. Beeswax (BW), with a melting point of 46–67 °C, primarily consisting of hydrocarbons, wax esters, free fatty acids and free fatty alcohols, was the product of Tianjin Guangfu Technology Co., Ltd. (Tianjin, China). Nile Red and Nile blue (>98% purity) were bought from Sigma Aldrich (St. Louis, MO, USA). Hydrochloric acid, sodium chloride and sucrose (analytical grade) were purchased from Beijing Chemical Works (Beijing, China). All the other food grade materials used were bought from a local supermarket, including edible salt (Jiangsu Salt Industry Group Co., Ltd., Nanjing, Jiangsu, China), white sugar (Taikoo Sugar Limited, Guangzhou, Gungdong, China), vinegar (Qianhe Condiment And Food Co., Ltd., Meishan, Sichuan, China), Kewpie mayonnaise (Beijing Q.p. FOODS Co., Ltd., Beijing, China), Hellmann mayonnaise (Shanghai Yinuo Food Co., Ltd., Shanghai, China) and eggs.

### 2.2. Preparation of Samples

HIPEs were developed by varying the contents of WPI, NaCl, and sucrose to investigate their effects on the HIPEs’ stability and properties. Then, mayonnaise-like HIPEs were prepared and compared with commercial mayonnaise to evaluate their viability as healthier alternatives.

#### 2.2.1. Preparation of Oleogels

Beeswax oleogels were made following the method described by Mohammad et al. [22] In brief, 4 wt% BW was blended with flaxseed oil using an IKA RCT basic heating magnetic stirrer (IKA-Werke GMBH and Co., Staufen, Germany). The mixture was stirred at 80 °C and 600 rpm for 10 min to achieve homogeneity, then promptly cooled to 25 °C with water.

#### 2.2.2. Preparation of HIPEs with Different WPI Contents

Following the modified method from Zhang et al. [23], HIPEs with 75 wt% oil phase were developed. In brief, HIPEs were prepared by dripping the oil phase to the aqueous phase and shearing at high speed. Different contents of WPI were dispersed in deionized water (3 wt%, 4 wt% and 5 wt% in the final HIPEs system) and left overnight for full hydration. Since mayonnaise is an acidic food (pH = 3.6~4.0), pH of the WPI dispersions was adjusted to 4.0 with HCl solution [24].

HIPEs were developed with a fixed aqueous to oil phase ratio (75:25). The beeswax oleogel was melted at 55 °C and added to the WPI dispersions drop by drop under high-speed shearing (55 °C, 11,000 rpm for 8 min) using an T18 Ultra-Turrax mixer (IKA-Werke GMBH and Co., Staufen, Germany). The prepared samples were promptly cooled to 25 °C. HIPEs containing 3%, 4% and 5 wt% WPI were termed WPI-3%, WPI-4% and WPI-5%, respectively. 

#### 2.2.3. Preparation of HIPEs with Different NaCl Contents

During the sample preparation process of Section 2.2.2, 4 wt% WPI was found to be sufficient for the effective forming of stable HIPEs. Before preparation, 4 wt% WPI dispersions with different NaCl contents (0 mM, 50 mM, 125 mM and 200 mM) were prepared (pH = 4.0). HIPEs were developed using a one-step homogenization process, as described in Section 2.2.2. The HIPEs with 0, 50, 125, 200 mM NaCl were termed NaCl-0, NaCl-50, NaCl-125 and NaCl-200, respectively.

#### 2.2.4. Preparation of HIPEs with Different Sucrose Contents

Before preparation, 4 wt% WPI dispersions with different sucrose contents (0, 2, 4 and 6 wt%) were prepared (pH = 4.0). HIPEs were developed using a one-step homogenization process, as described in Section 2.2.2. The HIPEs with 0, 2, 4 and 6 wt% sucrose were termed sucrose-0%, sucrose-2%, sucrose-4% and sucrose-6%, respectively.

#### 2.2.5. Preparation of Mayonnaise-Like HIPEs 

Simulated mayonnaise HIPEs with two different salt contents were prepared. One had 1% salt added (termed WE-1.0%), which is close to the additive amount of commercial mayonnaise. The other had 0.3% salt added (termed WE-0.3%) for salt reduction, so that the simulated mayonnaise had the characteristics of both low cholesterol and low salt content (Table 1). Before preparation, WPI, edible salt and white sugar were added to prepare the water phase (pH = 4.0). Simulated mayonnaise HIPEs were developed using a one-step homogenization process, as described in Section 2.2.2. 

To better simulate mayonnaise, egg yolk was used as emulsifier instead of WPI. In addition, 4 wt% BW oleogels constituted the oil phase. The separated egg yolk was mixed with water, salt, sugar and vinegar (Table 1) to prepare the water phase by stirring for 1 h. The mayonnaise samples (termed YE) were prepared using a one-step homogenization method by referring to the procedure described in Section 2.2.2. Finally, mayonnaise samples (WE-1.0%, WE-0.3%, YE) were compared with Kewpie mayonnaise (termed QB) and Hellmann mayonnaise (termed HLM) from the market.

### 2.3. Characterization of the Samples

#### 2.3.1. Particle Size

Referring to our previous method [25], droplet size of the samples was evaluated with a particle size analyzer (LS 13 230, Beckman Coulter Life Sciences, Indianapolis, IN, USA). HIPEs were diluted 100 times with deionized water to minimize the impact of multiple scattering. The relative refractive index (RI) of the oil phase was set at 1.470, and the RI of the water phase was set at 1.330. Each measurement was carried out in triplicate at 25 °C. Volume weighed mean diameter was determined to represent the mean particle size. 

#### 2.3.2. ζ-Potential

ζ-potential of the samples was evaluated using a dynamic light-scattering device (Nano-ZS90, Malvern Instruments Ltd., Worcestershire, UK). The samples were also diluted 100 times. The diluted samples were added in sample cells, and every measurement was carried out in triplicate at 25 °C.

#### 2.3.3. Microstructural Observation

Confocal laser scanning microscopy (CLSM) observation of the samples was carried out with an inverted confocal microscope (Zeiss-780, Inc., Jena, Germany). Nile Red and Nile Blue (1 mg/mL) were applied to stain the oil phase and protein prior to testing. The wavelengths of excitation and emission were selected as 488 nm and 633 nm, respectively. A 63× objective magnification was used to capture the images.

#### 2.3.4. Physical Stability

Physical stability of the HIPEs was evaluated with a LUMiSizer (L.U.M. GmbH, Berlin, Germany). Samples were centrifuged at 4000 rpm for a duration of 7200 s at 25 °C. Throughout the centrifugation process, the light transmittance intensity of the HIPEs was tested every 10 s. Changes in light transmittance were monitored, and the integral transmission (%) over time was recorded. 

#### 2.3.5. Oil Holding Capacity (OHC) and Water Holding Capacity (WHC)

OHC and WHC were tested to evaluate the stability of emulsions [26]. Each sample was added into a pre-weighted centrifuge tube (W_0_). Subsequently, the tube containing sample (W_1_) was centrifuged at 10,000 r/min for 30 min. After centrifugation, the released oil at the top of the sample was removed, and the remaining tube was weighed (W_2_). The released water (at the bottom) was removed, and the remaining tube was weighed (W_3_). The OHC (%) and WHC (%) were calculated with the following equation, and the reported values were the averages of three replicates.
(1)$$\mathrm{OHC}=(1-\frac{W_{1}-W_{2}}{(W_{1}-W_{0})\times \mathrm{Oil}\;\mathrm{proportion}})\times 100\%$$
(2)$$\mathrm{WHC}=(1-\frac{W_{2}-W_{3}}{(W_{1}-W_{0})\times \mathrm{Water}\;\mathrm{proportion}})\times 100\%$$

#### 2.3.6. Rheological Characterization

Rheological tests were conducted using a HAAKE IQ AIR rheometer (Thermo Scientific Inc., Karlsruhe, Germany) with a steel parallel plate (35 mm diameter and 1 mm spacing) in reference to the methodology outlined by Zhang et al. [27] Samples were equilibrated at 25 °C for 3 min before tests. To determine the samples’ linear viscoelastic region (LVR) between 0.01 and 100% (1 Hz), strain sweeps were performed. The steady shear viscosities were carried out at shear rates from 0.1 s^−1^ to 10 s^−1^. 

#### 2.3.7. Tribological Characterization

Following the procedure described previously [15], the samples were tested with a strain-controlled rheometer (ARES G2, TA Instrument, Crawley, UK) equipped with a three-ball geometry. The setup included a polydimethylsiloxane (PDMS) plate and a three-ball geometry to mimic the tongue surface and palate. To better replicate oral conditions, the PDMS plate was soaked in artificial saliva for 2 h. The measurement was conducted under a force of 2 N at 37 °C. Friction coefficients were recorded at an entrainment speed ranging from 0.01 to 100 mm/s. 

### 2.4. Statistical Analysis 

Statistical analysis was performed with Origin 2023 and SPSS 20.0 package. All of the tests were repeated three times. Differences between samples and effects of treatments were assessed using one-way analysis of variance (ANOVA). A significance level of *p* < 0.05 was used throughout the study.

## 3. Results and Discussion

### 3.1. Characterization of Oleogel-in-Water HIPEs

#### 3.1.1. Appearance and Microstructural Observation

The HIPEs could be prepared under acidic condition (pH = 4), which could better simulate mayonnaise (Figure 1). All the samples presented a smooth and pale-yellow appearance. The HIPEs displayed a soft-solid property and self-supporting gel-like structure. This gel-like structure was mainly due to the high volume of the internal phase and dense packing of the oil droplets [15]. There was no obvious difference among HIPEs with different WPI concentrations, and they all presented a semi-solid state (Figure 1A). Figure 1B shows that the fluidity of HIPEs was enhanced with higher NaCl content, which indicated that the addition of NaCl weakened the gel-like structure of HIPEs. Figure 1C indicates that the HIPEs containing 6% sucrose showed stronger self-supporting structures, comparing with the HIPEs without sucrose. This behavior suggested that addition of sucrose could enhance the structural strength of HIPEs.

The CLSM images revealed that oil droplets (shown in green) were distinctly dispersed in the aqueous phase containing WPI (shown in red). Consequently, the oil-in-water nature of the samples was confirmed. The close packing of oil droplets was a typical characteristic of HIPEs [28]. As WPI content increased, droplet size of the HIPEs decreased and the size distribution became more uniform (Figure 1A). When more WPI was added, more protein molecules were absorbed at the interface, and the oil drop was wrapped by a thicker protein film. WPI could lower the interfacial tension between the oil and water phases, which facilitated the formation of smaller droplets during the emulsification process [29]. When the addition of NaCl was increased, the droplet size grew, droplet coalescence occurred and many large droplets appeared (Figure 1B). When more NaCl was added, the ions dispersed in the water phase and could neutralize the charges on the droplet surfaces. This process, known as electrostatic shielding, reduced the repulsive forces between droplets, making them more prone to coalescence [30]. In Figure 1C, when the content of sucrose was increased, the droplet size was decreased gradually. A higher content of WPI added to the HIPEs could increase the viscosity of the water phase. This elevated viscosity helped to prevent droplet movement and collision, thus reducing the likelihood of coalescence. This resulted in smaller and more uniformly sized droplets [31].

#### 3.1.2. Droplet Characteristics

Droplet size of HIPEs with different WPI contents ranged from 10 μm to 15 μm, with high surface charge ranging from 30 mV to 35 mV (Figure 2A). The increase in WPI content (3–5%) led to a significant decline in particle size (*p* < 0.05), which was in line with the results of Figure 1A. Gao et al. [32] have found that WPI had good emulsifying capacity, and it provided emulsions with favorable stability via steric and electrostatic repulsion. Therefore, more WPI worked as emulsifier to promote the formation of smaller droplets. The surface charge was primarily attributed to the absorption of WPI, which could disperse droplets through electrostatic repulsion and helped maintain the stability of HIPEs [33]. No significant difference in ζ-potential was observed by increasing the WPI concentration (*p* > 0.05), suggesting fully cover of the interface by WPI.

When more NaCl was added, particle size of HIPEs was increased and ζ-potential value was decreased significantly (*p* < 0.05) (Figure 2B). The droplet size of HIPEs without added NaCl was the smallest (14.36 ± 0.47 μm), while the droplet size with an ionic strength of 200 mM was the largest (25.21 ± 1.99 μm). Meanwhile, the ζ-potential of HIPEs was decreased from 30.53 ± 0.67 mV (0 mM NaCl) to 20.23 ± 0.55 mV (200 mM NaCl). HIPEs stabilized by WPI were sensitive to the change in ionic strength. This result was closely linked to the electrostatic shielding effect of salt. When more NaCl was added, the shielding effect on the negative charge of WPI was enhanced, resulting in a lower net surface charge of droplets [30]. NaCl shielded the surface charge of WPI and reduced the interaction among WPI at the interface and WPI in water. Thus, the emulsification efficiency of WPI was weakened, leading to an increase in particle size [34].

With the increase in sucrose content from 0% to 6%, particle size of HIPEs showed a downward trend (Figure 2C). The HIPEs without sucrose had the largest particle size of 14.30 ± 0.15 μm, while particle size of the HIPEs containing 6% sucrose was the smallest (6.22 ± 0.21 μm). The increase in sucrose content led to a decrease of approximately 8.08 μm in particle size. Meanwhile, the addition of sucrose decreased the ζ-potential of HIPEs from 34.05 ± 0.94 mV to 29.20 ± 0.52 mV. The significant decrease (*p* < 0.05) in particle size was mainly because the addition of sucrose could inhibit the flocculation of droplets [35], which helped to form smaller droplets. Huck-Iriart et al. [36] found that the size distribution of sodium caseinate-stabilized emulsions without sucrose had double peaks. However, only a single peak was observed in the emulsion with 20 wt% sucrose, and the particle size became smaller. The impact of sucrose could be interpreted in different ways. It might establish hydrogen bonds between the hydroxyl groups of sucrose and the carboxylic groups of protein, or it could function as a solvent of protein modifying interactions among protein molecules [34]. The decline of ζ-potential was because more sucrose molecules would wrap the charged polar groups of WPI and led to the reduction of net charge. Similar results were also found by Li et al. [37], who noted that adding sucrose could reduce ζ-potential of mayonnaise-like Pickering emulsion stabilized by pea protein isolate microgels from 23.6 ± 0.6 mV to 11.0 ± 0.2 mV. 

#### 3.1.3. Physical Stability

In Figure 3, the increased light transmission (the right end of the scale bar) revealed the movement of oil droplets to the upper layer, which typically resulted in creaming at the bottom of the samples. All HIPEs exhibited greater light transmission near the bottom, indicating that oil droplets migrated upwards and some phase separation occurred [38]. The transmission intensity fluctuated significantly less in the HIPEs containing 4% and 5% WPI compared to that in HIPEs containing 3% WPI. The HIPEs with 5% BW exhibited the smallest variation, indicating that increasing WPI content enhanced the stability of HIPEs. In addition, as the WPI concentration was increased, the particle size of HIPEs was gradually decreased, and the sedimentation rate of droplets declined, which helped to improve the stability of HIPEs. Figure 3B shows that higher NaCl concentration significantly increased light transmission intensity, suggesting the lower stability of HIPEs. Furthermore, there was an obvious peak in the upper part of the tube of 125 mM and 200 mM samples (the left end of the scale bar), suggesting that some liquid oil was released after centrifugation [39]. In the HIPEs with added sucrose, there was lower transmission intensity and a narrowed destabilizing area, indicating that sucrose had positive effects on the stability of HIPEs. The addition of sucrose increased the aqueous phase viscosity, thereby reducing the possibility of droplet movement and collision. It should be pointed out that the above results were obtained under accelerated conditions. Actually, no visual phase separation was observed under quiescent conditions throughout the study.

#### 3.1.4. Oil Holding Capacity (OHC) and Water Holding Capacity (WHC)

As the WPI concentration increased, OHC and WHC of HIPEs were increased significantly (*p* < 0.05) (Figure 4A,B). HIPEs with 4% and 5% WPI presented much higher OHC (>95%), suggesting improved stability. When the WPI content was increased from 3% to 5%, WHC of HIPEs was increased from 75.06 ± 0.45% to 86.05 ± 0.32%. The formation of thicker film by protein at the interface significantly reduced interfacial tension, which was beneficial for emulsion stability [40]. In addition, HIPEs with higher WPI concentration had smaller particle size, which helped to improve the kinetic stability of HIPEs. OHC and WHC of HIPEs showed a gradual decrease when NaCl content was increased (Figure 4C,D). HIPEs without NaCl had the highest OHC (99.75 ± 0.11%) and WHC (86.05 ± 0.32%). By contrast, HIPEs with 200 mM NaCl had the lowest values of OHC (73.91 ± 1.25%) and WHC (81.79 ± 0.18%), indicating that NaCl reduced the centrifugal stability of the HIPE systems. In Figure 4E,F, HIPEs with different sucrose contents maintained a high value of OHC, around 99%. In addition, WHC of HIPEs was increased when more sucrose was added, from 86.05 ± 0.32% to 92.34 ± 0.01%, suggesting that the addition of sucrose had positive effects on the stability of the HIPEs. Kim et al. [41] also found that sucrose enhanced the resistance and stability of emulsions against coalescence and flocculation.

#### 3.1.5. Rheological Properties

The resistance of HIPEs to deformation was measured via rheological analysis via strain sweeps (Figure 5A). Within LVR (<1% strain), G′ dominated over G″ in all samples, suggesting the elastic properties of the HIPEs. This originated from the BW oleogels network structures in the oil phase, and the network formed by the closely packed oil droplets. Additionally, the entanglement networks between adsorbed and non-adsorbed protein molecules also contributed to the highly elastic properties of the HIPEs [42]. At the strain beyond LVR, G′ and G″ decreased, and G′ was gradually overtaken by G″, suggesting that increased stress levels could irreversibly disrupt the structure of the HIPEs. This change in modulus showed the transition of the gel structures from elastic-dominated properties to viscous-dominated properties [43]. In addition, both G′ and G″ increased with the rise in WPI content, suggesting that WPI in HIPEs could enhance their viscoelasticity. Chang et al. [44] proposed that a high concentration of soy protein isolate in HIPEs facilitated the formation of a stronger viscoelastic layer at the interface. Similarly, the increase in WPI content had a positive impact on the formation of viscoelastic layer at the oil–water interface. In addition, excessive WPI remaining in the water phase filled the gaps between droplets, which could function as a thickening agent [45]. In Figure 5B, all the HIPEs showed rheological properties of weak gels. After the addition of NaCl, G′ and G″ of the HIPEs were lower than those of HIPEs without NaCl, indicating that NaCl weakened the viscoelastic characteristics of HIPEs. As sucrose concentration increased, G′ and G″ of HIPEs increased under the small amplitude oscillatory shear (Figure 5C). The experimental results indicated that adding sucrose could improve the viscoelastic characteristics of HIPEs.

Figure 5D shows that as the shear rate increased, all HIPEs exhibited a decrease in apparent shear viscosity. The viscosity was increased when more WPI was added. Since more WPI molecules were extensively adsorbed onto the oil-water interface, they formed a denser and more stable interfacial layer. This led to stronger structures and improved shear resistance. All HIPEs had a shear-thinning behavior, which was attributed to the deformation and disruption of highly densely packed oil droplets. At lower shear rates, the HIPEs exhibited higher viscosity due to the intact networks within the oil phase, which offered increased resistance against shearing. However, as the shear rate continued to increase, the viscosity decreased almost to its minimum value. The increased shear rate led to more frequent collisions between oil droplets, causing aggregation and permanent deformation of the structures [46]. In HIPEs, the networks formed by densely packed oil droplets, with or without interfacial interactions, tended to break down under higher shearing. Secondly, the oleogel formed by BW was also susceptible to shear forces [23]. Figure 5E reveals that viscosity of HIPEs with NaCl was lower compared to those without NaCl, especially at high shear rate. When the shear rate exceeded 2 s^−1^, the viscosity values of HIPEs with NaCl (50 mM, 125 mM and 200 mM) fell to ~13 Pa·s. This result suggested that the addition of NaCl reduced the viscosity of HIPEs stabilized by WPI. It was also noted that the HIPE with 200 mM NaCl showed the lowest viscosity, possibly because the system was more sensitive to shearing. Secondly, droplets of bigger size might also contributed to lower viscosity [47]. During the whole test (0.1~10 s^−1^), the viscosity of HIPEs containing sucrose was always higher than that of HIPEs without sucrose (Figure 5F). For example, at the shear rate of 0.1 s^−1^, the viscosity of HIPEs with sucrose content of 0%, 2%, 4% and 6% were 556 Pa·s, 1485 Pa·s, 2000 Pa·s and 1650 Pa·s. The increase in viscosity could be due to the dispersion of sucrose in the water phase, leading to an elevation in the viscosity of the continuous phase. Junqueira et al. [48] found that sucrose promoted steric stability and increased the viscosity of the emulsions by hindering the mobility and aggregation of droplets. Additionally, the addition of sucrose effectively reduced the average particle size of the HIPEs, which also contributed to higher viscosity [49].

### 3.2. Characterization of Mayonnaise Samples

#### 3.2.1. Morphology

Visual observation revealed the similar appearance of mayonnaise-like emulsions (WE-1.0%, WE-0.3%, YE) and commercial mayonnaise samples (QB, HLM) (Figure 6). They all had similar plasticity, a delicate and smooth texture, and spreadability. The self-supporting nature and smoothness of commercial mayonnaise samples were superior to those of oleogel-based mayonnaise samples. This could be attributed to the presence of thickeners, such as xanthan gum, in commercial mayonnaise products [50]. CLSM images confirmed the formation of O/W emulsions, as the oil droplets (in green) were dispersed in the aqueous phase containing proteins (in red). The droplets in WE-1.0% had an uneven distribution, and there was evident droplet aggregation. YE, QB, and HLM exhibited a higher degree of droplet aggregation and smaller droplet size. This was probably because the emulsifying property of egg yolks was better than that of WPI. It was concluded that smaller droplet size contributed to better taste and mouthfeel of the mayonnaise and provided higher stability [51].

#### 3.2.2. Rheological Properties

Strain sweep tests were conducted to assess the changes in viscoelastic behavior of the mayonnaise samples under deformation (Figure 7A). Within the LVR region, all samples exhibited G′ higher than G″, revealing a dominant elastic response. Compared to the two commercial mayonnaise samples, the three oleogel-based mayonnaise samples had relatively higher G′ and G″, which was related to the gelation of the internal phase. According to our previous study, it was found that adding BW into the oil phase could strengthen the internal network of the emulsions and enhance their resistance to deformation [25]. Similar results were also reported in a literature study [23], where BW crystallization within the oil phase trapped the liquid oil, resulting in higher elasticity of the emulsions. G′ of YE was higher than that of WE-0.3% and WE-1.0%, which could be attributed to the good emulsifying ability of egg yolk. Literature studies indicated that the viscoelasticity of mayonnaise mainly came from the network structures formed between lipoproteins, which adsorbed around the surface of oil droplets and provided a viscoelastic interface for mayonnaise [52]. As the strain applied on the samples was increased, G″ gradually dominated G′, indicating the structural breakdown of the samples.

All the mayonnaise samples displayed shear-thinning behavior as the viscosity values fell with shear rate (Figure 7B), which was attributed to the deformation and destruction of the clusters or aggregates of droplets [53]. At the shear rate of 0.1 s^−1^, the viscosity values of three oleogel-based mayonnaise samples were very close (around 1000 Pa·s). In comparison, the viscosity values of commercial mayonnaise QB and HLM were much lower (~550 Pa·s and 260 Pa·s). This difference was due to the higher content of WPI or egg yolk (4%) in the laboratory-prepared samples compared to the emulsifier content in commercial products, which was about 2%. When the shear rate was increased to 10 s^−1^, the viscosity of QB and HLM turned to about 16 Pa·s, while the viscosity of the other three samples decreased more rapidly, reaching approximately 7 Pa·s. This result indicated that, at high shear rates, commercial mayonnaise products exhibited better shear resistance, possibly due to the addition of food additives such as xanthan gum.

#### 3.2.3. Tribological Properties

Tribological tests could offer insights into textural attributes, such as smoothness and creaminess of food, during oral processing [54]. Stribeck curves revealed the lubricating behavior of the samples (Figure 8). The Stribeck curve was typically classified into three regimes: the boundary regime, in which friction was determined by the properties of the contact surface rather than the entrainment speed; the mixed regime, in which the fluid began to be entrained into surfaces and formed a lubricating film, reducing frictions; and the hydrodynamic lubrication regime, in which frictions increased as the bulk dominated [55]. All the emulsions followed the shape of classical Stribeck curves. At the beginning of the test (sliding speed < 0.03 mm/s), samples had a constant friction coefficient (μ) value within the boundary lubrication regime. At this stage, the friction behavior was mainly dominated by surface asperities and interfacial film. Compared with oleogel-based mayonnaises, HLM and QB showed higher boundary lubrication regime and the higher friction coefficient (μ ≈ 0.4). This was possibly due to the addition of various food additives (in commercial products), which led to the increase in surface asperities and reduced lubrication. The boundary regime of mayonnaise samples was narrow, which was attributed to the presence of oil that facilitated the formation of lubrication film [15]. As sliding speed increased (from 0.03 to 1 mm/s), friction coefficients of all the samples declined, reaching the mixed or intermediate regimes. In this regime, more emulsion and oil were absorbed into the region between the sliding surfaces, leading to a decrease in μ. Due to the hydrophobicity of PDMS, it was easy for oil to form a sticky and continuous layer on the surface of PDMS, which was the reason for the low μ value. When sliding speed increased, structural damage occurred in BW oleogels, and the oil phase was diffused to form lubricating films, leading to a rapid decrease in friction coefficient. As sliding speed was over 1 mm/s, the friction coefficients of WE-0.3%, WE-1.0% and QB rose, suggesting the transition of tribological behavior into the hydrodynamic regime. When more emulsions entrained into the friction surfaces, the lubrication effect became determined by the bulk rheology of the lubricant [56]. Tightly packed droplets and higher viscosity contributed to increased friction coefficients. In general, the Stribeck curves of WE-0.3% and WE-1.0% were similar to QB, and the Stribeck curve of YE was similar to HLM. It was found that WE-0.3% and WE-1.0% had lower friction values, while QB displayed higher friction values. This indicated that WE-0.3% and WE-1.0% had better lubricating properties and smoother perception, which might be attributed to the nature of the oleogel, as wax-based materials tended to reduce friction coefficient [57]. In general, oleogel-based mayonnaises could be designed to have similar rheological and tribological properties as commercial mayonnaises, revealing the great potential of oleogel-based HIPEs for the development of mayonnaise products.

## 4. Conclusions

The current study developed oleogel-in-water high internal phase emulsions stabilized using WPI and BW oleogels. The results revealed that, at pH 4, the HIPEs could be stabilized with only 4% and 5% WPI. The increase in WPI concentration could effectively increase the viscoelasticity and stability of HIPEs. The addition of NaCl weakened the structures and stability of HIPEs. The addition of sucrose could reduce the droplet size of HIPEs and increase its stability. Finally, two simulated mayonnaises WE-0.3% (low salt) and WE-1.0% were prepared with 4% WPI and 2% sucrose at pH 4. The results concluded that appearance, rheological and tribological properties of commercial mayonnaises could be mimicked by oleogel-based HIPEs, which also revealed the resembled creaminess and smoothness properties. Overall, the findings contribute valuable insights into the feasibility of using oleogel-in-water HIPEs as a promising emulsion system for simulating mayonnaise.

## Figures and Tables

**Figure 1 foods-13-02738-f001:**
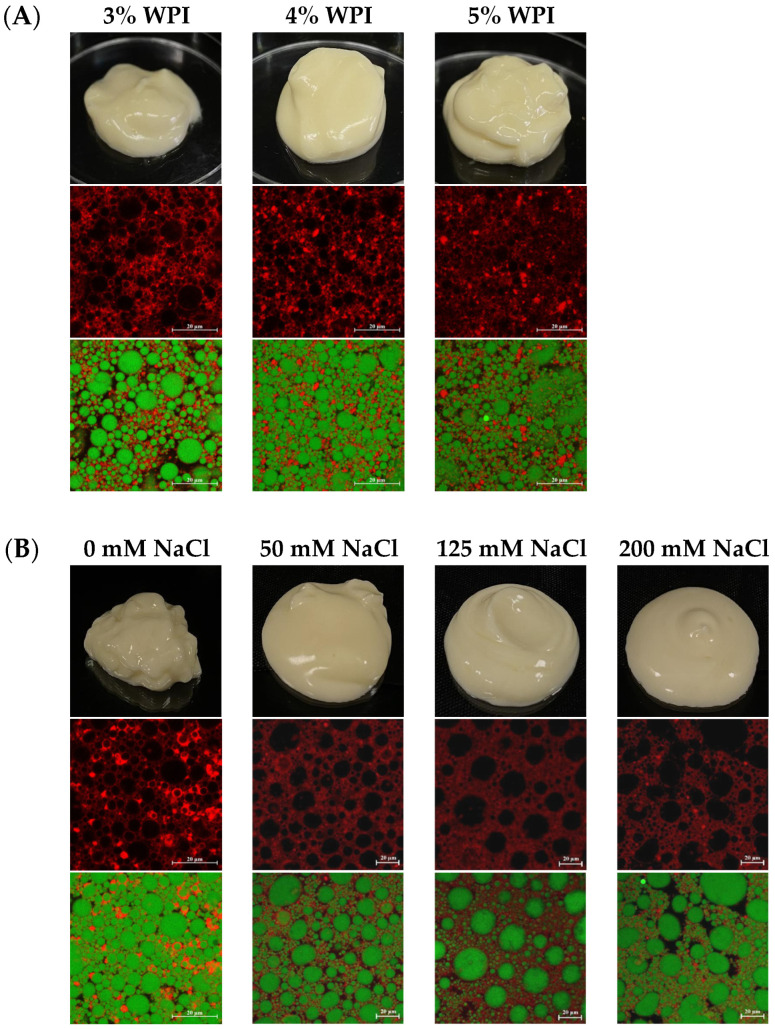
Appearance and CLSM images of HIPEs with different WPI contents (**A**), NaCl contents (**B**) and sucrose contents (**C**). The scale bar is 20 μm.

**Figure 2 foods-13-02738-f002:**
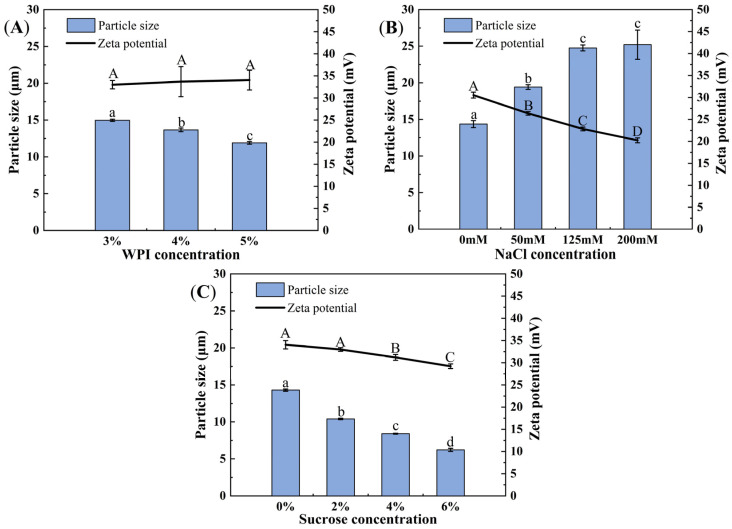
Particle size and ζ-potential of HIPEs with different WPI contents (**A**), NaCl contents (**B**) and sucrose contents (**C**). Different letters above bars indicate significant differences of particle size and ζ-potential (*p* < 0.05). Error bars represent standard errors.

**Figure 3 foods-13-02738-f003:**
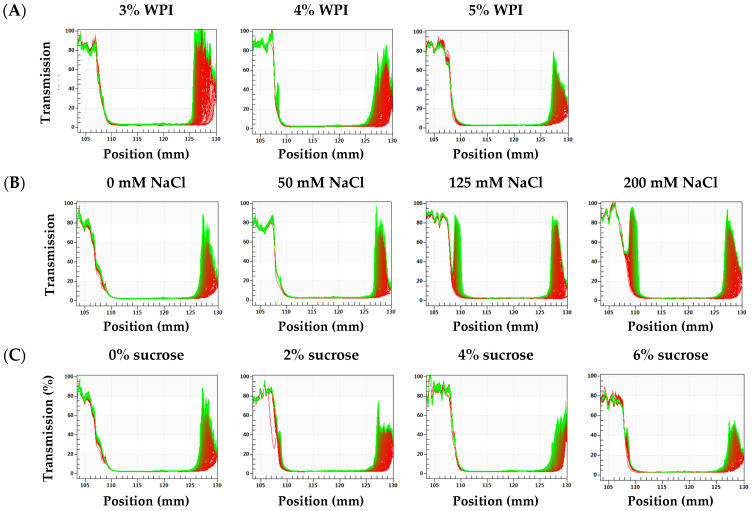
LUMiSizer profiles of HIPEs with different WPI contents (**A**), NaCl contents (**B**) and sucrose contents (**C**).

**Figure 4 foods-13-02738-f004:**
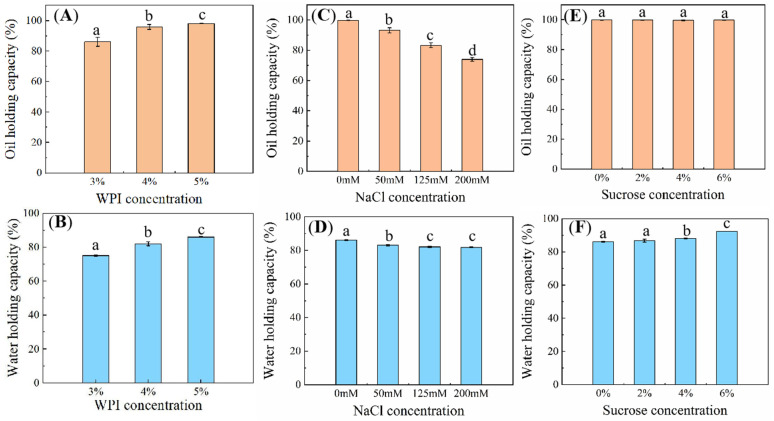
Oil holding capacity of HIPEs with different WPI contents (**A**), NaCl contents (**C**) and sucrose contents (**E**). Water holding capacity of HIPEs with different WPI contents (**B**), NaCl contents (**D**) and sucrose contents (**F**). Different letters above bars indicate significant differences for oil holding capacity and water holding capacity (*p* < 0.05). Error bars represent standard errors.

**Figure 5 foods-13-02738-f005:**
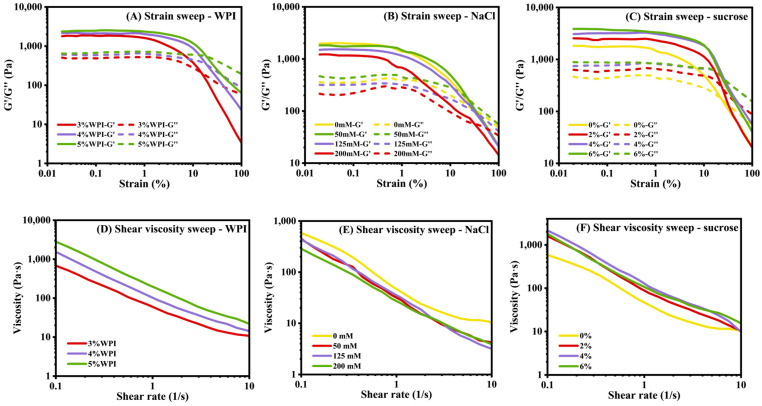
Rheological properties of HIPEs with different WPI contents, NaCl contents and sucrose contents.

**Figure 6 foods-13-02738-f006:**
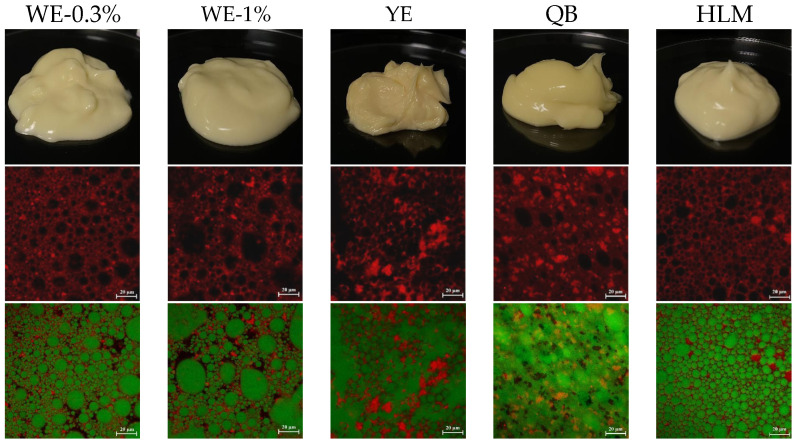
Appearance and CLSM images of different mayonnaise samples. The scale bar is 20 μm.

**Figure 7 foods-13-02738-f007:**
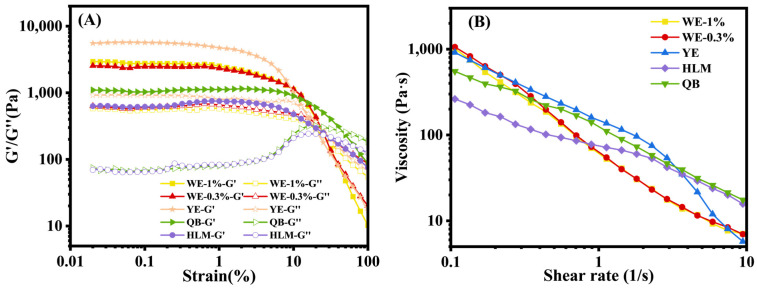
Rheological properties of different mayonnaise samples. ((**A**) Strain sweep, (**B**) Shear viscosity sweep).

**Figure 8 foods-13-02738-f008:**
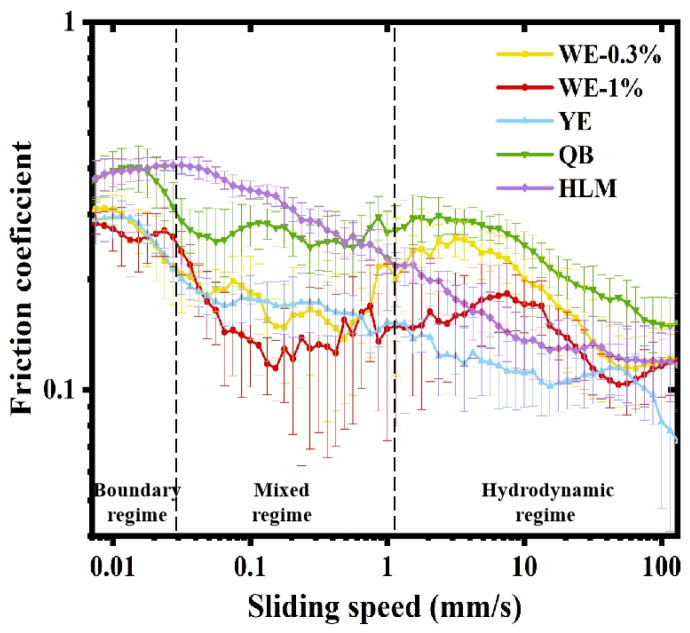
Stribeck curves were obtained for different mayonnaise samples.

**Table 1 foods-13-02738-t001:** Ingredients list of mayonnaise samples.

Sample	WPI	Yolk	Oil Phase	Edible Salt	White Sugar	Vinegar	Others
WE-0.3%	4.0 wt%	/	75 wt%	0.3 wt%	2.0 wt%	/	water
WE-1.0%	4.0 wt%	/	75 wt%	1.0 wt%	2.0 wt%	/	water
YE	/	4.0 wt%	75 wt%	1.0 wt%	2.0 wt%	10 wt%	water
QB	/	2.0 wt%	75 wt%	1.5 wt%	2.5 wt%	10 wt%	food additives and water
HLM	/	2.5 wt%	75 wt%	1.0 wt%	2.0 wt%	10 wt%	food additives and water

## Data Availability

The original contributions presented in the study are included in the article; further inquiries can be directed to the corresponding author.

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
