# Peer review of "Development of Oleogel-in-Water High Internal Phase Emulsions with Improved Physicochemical Stability and Their Application in Mayonnaise"

_foods, 2024, doi:10.3390/foods13172738_

Round 1
Reviewer 1 Report
Comments and Suggestions for Authors
The paper is very interesting and gives some important knowledge on the fast developing field of "alternative" products. However, some comments need to be addressed:
P1, L32: Please remove the word "oil" at the end of the sentence.
P1, L40: Please correct the word "literatures" to "literature".
In the Introduction section, you explain the possible effects of all the components used in mayonnaise production on its physical properties. The only components missing are bees wax and flax seed oil. Please add a sentence or two on why these components are used in mayonnaise formulations and what are its positive and negative sides.
P3, L110: Please list the equipment used for blending.
I am somewhat confused about the total number of samples produced for analysis. Namely, in Table 1, you list that you had 5 different samples. However, in the R&D section there are numerous results on the effect of WPI, sucrose and NaCl. Are the samples listed in Table 1 only the base samples, to which different concentrations of NaCl, sucrose and WPI were added? Also, in that table it says that there was only one WPI level tested, 3 sucrose levels and 3 NaCl, yet in R&D there are more. Please provide a Table with a complete list of samples with clear differences among their composition.
Another question which arises, if you varied WPI, NaCl and sucrose, why didn't you use experiment design? E.g. Box Behnken design would reduce the number of needed experiments and produce clear and firm results by statistical analysis (ANOVA). Also, optimization of the mixture would be possible. Please explain why you used the "trial and error" method instead of any type of experiment design.
P4, Particle size analysis: Equation 1 is not necessary. Please remove it, it is enough just to state that volume weighed mean was determined.
P4, L166: Why were the samples diluted and did it have an effect on the zeta potential?
P7, discussion regarding Fig.2. In which zeta potential range are emulsions considered stable and are all of the emulsions you produced in that range? Namely, a drop in zeta potential does not necessarily mean that emulsions lose their stability.
Comments on the Quality of English LanguageSome minor corrections required.
Reviewer 2 Report
Comments and Suggestions for Authors
The research of the current manuscript concerns the evaluation of characteristics an oleogel-in-water high internal phase emulsions on mayonnaise.
In my opinion the research is almost interesting and actual. However, the manuscript needs important major revision.
Introduction.
I don’t agree with the author about the negative health effect of the egg yolk. the authors must not demonize a food that has excellent nutritional and health properties that food researchers should know about. In this way, incorrect and dangerous information is given. what is meant by overconsumption???
Please describe properly the aim and the development of this work for a better understanding of the successive paragraphs.
Material and methods
This part should be deeply improved. Authors should specify what was carried out in a proper manner reporting what was studied during the first and the second phase of the experimentation.
For example, where and why the commercial mayonnaises were used is not absolutely clear.
Results and Discussion
Authors should deeply improve this part because it lacks in part the statistical analysis. In fact, the comparison among mayonnaises was not supported by statistical analysis. In this regard, I have to underline the importance of the validation of the data gathered by applying a proper statistical analysis. Why did the authors apply it only on the first part of the experimentation.
Furthermore, the significance of data must be underlined throughout the text. Are data significant or not significant different??? p value???
In my opinion, this research is penalized by the fact that
1) the sensory analysis was not carried out and I do think that is not acceptable when a very important modification is done on a food product, especially a very common commercial product like mayonnaise, an experts’ judgment followed by a consumer opinion should be required and shown. The instrumental analysis carried out by the authors to replace the sensory analyses were not sufficient to cover alle the sensory aspects which the different oleogel-in-water and salt concentrations could modify such as the smell and the taste properties.
Authors focused only on the visual evaluation but the mayonnaise gives particular smell, taste and texture sensations which are fundamental for the costumer choice. I suppose that the authors want to evaluate the oleogel in water with a view to commercial production of a low-cholesterol mayonnaise. Otherwise, I don't see the use of this experimentation. Therefore, an in-depth sensory analysis that evaluates all the human senses involved and not just sight and texture, must be conducted. Alternatively, an instrumental evaluation of volatile compounds which determine the odorous characteristics should be given.
2) A shelf life should be carried out; in fact, the oleogel-in-water rheologic properties should be maintained for the entire shelf life of the mayonnaise in order to avoid unpleasant effects of phases separation. Thus, their long lasting rheologic capabilities should be demonstrated.
According to all my observations, I think that this manuscript will be suitable for Foods only if data of a shelf life and a sensory analysis will be included.
